# Liver Injury and Regeneration: Current Understanding, New Approaches, and Future Perspectives

**DOI:** 10.3390/cells12172129

**Published:** 2023-08-22

**Authors:** Shainan Hora, Torsten Wuestefeld

**Affiliations:** 1Genome Institute of Singapore (GIS), Agency for Science, Technology and Research (A*STAR), 60 Biopolis Street, Genome, Singapore 138672, Singapore; wustefeldt@gis.a-star.edu.sg; 2National Cancer Centre Singapore, Singapore 168583, Singapore; 3School of Biological Science, Nanyang Technological University, Singapore 637551, Singapore

**Keywords:** liver injury, liver regeneration, liver pathologies, hepatocytes, targeted therapeutics

## Abstract

The liver is a complex organ with the ability to regenerate itself in response to injury. However, several factors can contribute to liver damage beyond repair. Liver injury can be caused by viral infections, alcoholic liver disease, non-alcoholic steatohepatitis, and drug-induced liver injury. Understanding the cellular and molecular mechanisms involved in liver injury and regeneration is critical to developing effective therapies for liver diseases. Liver regeneration is a complex process that involves the interplay of various signaling pathways, cell types, and extracellular matrix components. The activation of quiescent hepatocytes that proliferate and restore the liver mass by upregulating genes involved in cell-cycle progression, DNA repair, and mitochondrial function; the proliferation and differentiation of progenitor cells, also known as oval cells, into hepatocytes that contribute to liver regeneration; and the recruitment of immune cells to release cytokines and angiogenic factors that promote or inhibit cell proliferation are some examples of the regenerative processes. Recent advances in the fields of gene editing, tissue engineering, stem cell differentiation, small interfering RNA-based therapies, and single-cell transcriptomics have paved a roadmap for future research into liver regeneration as well as for the identification of previously unknown cell types and gene expression patterns. In summary, liver injury and regeneration is a complex and dynamic process. A better understanding of the cellular and molecular mechanisms driving this phenomenon could lead to the development of new therapies for liver diseases and improve patient outcomes.

## 1. Introduction

The liver has a vital role in metabolic homeostasis, and its ability to utilize multiple mechanisms to maintain the liver-to-bodyweight ratio for proper functioning makes it an indispensable organ within the body. However, liver injury can result from various etiologies and can have significant consequences on hepatic function. Regardless of the underlying cause, liver injury commonly leads to inflammation, hepatocyte damage, and subsequent tissue remodeling. Liver injury, if left unchecked, can lead to chronic liver damage characterized by fibrosis and subsequently cirrhosis and/or hepatocellular carcinoma (HCC). In addition, the regenerative capacity of the liver is also compromised. This implies that liver transplantation is the only option for end-stage chronic liver disease; however, the demand for organs far outcompetes the supply. Even worse, in a future with an aged population and an increase in liver fibrosis, the relative organ supply is expected to shrink. This review sheds light on the recent advancements in liver regeneration and the mechanisms governing it and provides an overview of potential therapeutic strategies to address the underlying problem.

## 2. Uncovering the Molecular and Cellular Mechanisms of Liver Injury and Regeneration

The innate immune response serves as a first line of defense against invading pathogens by engaging germline-encoded pattern-recognition receptors (PRRs). These PRRs are expressed at the cell surface of innate immune cells, such as monocytes, macrophages, neutrophils, and epithelial cells, as well as adaptive immune system cells [1]. PRRs recognize conserved molecular structures called danger-associated molecular pattern molecules (DAMPs) or pathogen-associated molecular patterns (PAMPs). Upon hepatic injury, stimuli molecules, DAMPs or PAMPs, are released, activating Kupffer cells, among others. This activation leads to the release of pro-inflammatory cytokines, such as tumor necrosis factor-alpha (TNF-α) and interleukin-1 beta (IL-1β) [2]. In addition, resident macrophages or Kupffer cells and hepatocytic macrophages derived from circulating monocytes also play a crucial role in liver-related pathogenesis. This pro-inflammatory environment further perpetuates tissue injury and disrupts hepatic homeostasis.

Inflammation also aggravates liver etiology by damaging the liver tissue and promoting fibrosis by activating the hepatic stellate cells (HSCs) to deposit excessive amounts of extracellular matrix (ECM) proteins, including collagen, into the liver. HSCs constitute about 5–8% of cells in the liver and play a role in retinoid storage in their quiescent form [3]. Liver injury triggers these cells, which then transdifferentiate into myofibroblasts, that play a critical role in tissue repair and fibrosis. A recent study established the dual function of HSCs in hepatocarcinogenesis by analyzing mouse and human HSC subpopulations [4]. Quiescent and cytokine-producing HSCs protected hepatocytes from cell death and inhibited HCC development, while activated myofibroblastic HSCs promoted hepatocyte activation and tumor growth by altering the stiffness of the liver tissue and activating specific signaling pathways. The balance between these two HSC subpopulations also shifted during the progression of chronic liver disease, and an increased abundance of myofibroblastic HSCs was associated with a higher risk of developing HCC in patients.

Liver injury also initiates compensatory mechanisms for tissue regeneration. Hepatocytes go through three distinct phases during liver regeneration that involve several growth factors and key signaling pathways that contribute to the initiation and progression of hepatocytes. The distinct phases are: initiation (also known as priming), proliferation, and termination, which are governed by many signaling mechanisms and require a balance of pro- and anti-proliferating factors to prevent oncogenesis [5].

During the initial phase, hepatocytes exit their dormant state and begin the process of DNA synthesis. Two pro-inflammatory cytokines, interleukin 6 (IL-6) and tumor necrosis factor alpha (TNF-α), are known to be the main mediators involved in the process. Following innate immune response, stimulation by components such as lipopolysaccharide (LPS) or complement proteins upregulates TNF-α, which binds to its receptor TNF receptor 1 (TNFR1) in Kupffer cells and initiates a series of intracellular signal transduction events, leading to the activation of the NF-κB pathway through a process involving the IκB kinase (IKK) complex [6]. Upon activation, the IKK complex phosphorylates IκB proteins (inhibitors of NF-κB), leading to their degradation via the ubiquitin–proteasome pathway. This allows NF-κB dimers (typically p50 and p65 subunits) to be released and translocated from the cytoplasm to the nucleus, where they play a pivotal role in liver regeneration by integrating signals from the pro-inflammatory cytokines TNF-α and IL-6. NF-κB target genes also include growth factors (hepatocyte growth factor), anti-apoptotic proteins (Bcl-2 family members), and cell-cycle regulators (cyclin D1) [7].

In the hepatocytes, IL-6 binds to IL-6R, leading to the formation of a complex with glycoprotein 130 (gp130), which activates downstream signaling pathways. The IL-6R/gp130 complex activates the Janus kinase-signal transducer and activator of transcription (JAK-STAT) pathway. This results in the initiation of cell proliferation via Cyclin D1. TNF-α, on the other hand, stimulates hepatocyte c-Jun N-terminal kinase (JNK), which leads to the phosphorylation of c-Jun transcription factor in the nucleus. This activation induces the transcription of cyclin-dependent kinase 1 (CDK1), which promotes hepatocyte proliferation [6,7].

Thus, the activation of NF-κB and subsequent transcriptional changes contribute to hepatocyte proliferation by promoting the expression of genes required for cell-cycle progression. Additionally, NF-κB-mediated induction of anti-apoptotic proteins enhances hepatocyte survival in the context of the inflammatory microenvironment. However, it is important that NF-κB activity is tightly regulated to avoid prolonged inflammation, which could lead to tissue damage [6]. Subsequently, hepatocytes reach the G1 and early S phases of the cell cycle by several signals that activate growth factors and mitogens, such as hepatocyte growth factor (HGF) and epidermal growth factor (EGF). Cellular proliferation stops when the body mass ratio is attained and inhibitory molecules such as transforming growth factor-beta (TGF-β) restrict the rate and direction of liver regeneration [7] (Figure 1).

In addition to hepatocytes, another subpopulation in the liver known as liver progenitor cells (LPCs) is known to play a role in liver regeneration. Oval cells, the first liver progenitor cells, are a cell population capable of differentiating into hepatocytes and cholangiocytes, the major cell types of the liver [8]. Experimental models using lineage tracing studies have explored this differentiation potential in response to liver injury [9,10]. While one study showed Sox9-positive cells to be involved in regeneration after liver injury, another showed that Sox9-positive cells primarily contribute to oval cell proliferation and the formation of organoids but rarely give rise to hepatocytes in vivo. Thus, although Sox9, a transcription factor and cholangiocyte marker, has been studied in different stages of organ development, homeostasis, and regeneration, along with other markers, the role of LPCs in regeneration still requires further investigation.

Studies have also found that other non-parenchymal cell (NPC) populations, such as biliary epithelial cells (BECs), play a role in hepatocyte regeneration. A study showed that NPC-derived hepatocytes accounted for a substantial portion of the regenerated liver tissue. By utilizing a marker called cytokeratin 19 (CK19), BECs were identified as a crucial source of hepatocyte regeneration, particularly in cases of chronic liver injury [11,12].

While the precise mechanisms underlying different cell-type populations, their differentiation, and their interaction with the liver microenvironment are still being elucidated, advances in research have provided valuable insights into these processes. Further research is needed to fully understand their molecular and cellular characteristics and their potential clinical applications.

## 3. Acute vs. Chronic Liver Damage: How Predictive Is the Classic Partial Hepatectomy Model?

The extraordinary regenerative power of the liver was already known in ancient times, as described in the Greek tale of Prometheus, who was punished by the gods for stealing the fire. As punishment, Prometheus was chained to a rock and an eagle ate pieces of his liver every day, only so that the liver regenerated overnight, and the torture started all over again. However, a breakthrough for liver regeneration research and the foundation for a detailed understanding of the mechanism was presented by Higgins and Anderson in 1931 [13]. They showed that a two-thirds partial hepatectomy (PHx) induced a reliable liver regeneration response. This was possible as acute damage to the liver induced a synchronous response of the remaining cells. Many researchers used this model to dissect the underlying molecular changes of the regeneration process. Interestingly, if the surgical insult is limited to under 50%, the regeneration program mainly changes from hyperplasia to hypertrophy [14]. This indicates that more than one regenerative response is possible.

Despite the understanding of the molecular mechanism of liver regeneration, the PHx model does not reflect the most common situation of chronic liver disease. Chronic liver disease triggered by hepatitis viruses or non-alcoholic fatty liver disease (NAFLD) does not result in massive acute damage but a persistent insult. This triggers a continuous turnover of cells, hepatocyte death, and a continuous compensatory local regeneration. Over time, the regenerative capacity of the hepatocytes is exhausted, and a second line of defense is triggered, followed by the activation of a stem cell compartment and the oval cells of the liver. However, the stem cells are not able to fully compensate for the hepatocytic loss, leading to increased accumulation of fibrotic scar tissue and finally end-stage liver disease.

This raises the question whether the understanding gained from the acute liver damage model of PHx is sufficient to understand chronic liver disease and its progression to end-stage liver disease. As discussed earlier, pro-inflammatory signals such as TNF- α and IL-1β are important to trigger hepatocytes to enter the cell cycle and proliferate. However, chronic liver damage leads to chronic inflammation, which drives fibrosis, hepatocyte exhaustion, and disease progression [15,16]. In addition, continuous liver insult might disrupt the classic liver regeneration program seen in acute liver damage. Chronic liver damage also leads to the accumulation of senescent cells [17], cells that are permanently arrested in their cell cycle. Senescent hepatocytes therefore cannot contribute to liver regeneration and can even induce senescence in other cells via paracrine mechanisms. Over time, significant loss of hepatocytes is seen, which could contribute to inefficient liver regeneration. This explains the regenerative decline under chronic liver disease and aging. However, what was also found is that the regenerative program, when triggered, is not only strong but can initiate senescence escape, thereby contributing to liver cancer [18]. This highlights that by eliminating senescent hepatocytes the regenerative power of the liver under chronic liver disease and aging can be improved. This phenomenon has so far been observed in animal models [19,20,21].

Despite the decline in the regenerative response of hepatocytes under chronic liver damage and aging, serial hepatocyte transplantation experiments [22] have shown the potential of hepatocytes in contributing towards long-term liver regeneration. Additionally, functional genomics dissecting the biology, especially in the context of chronic liver damage, have helped to identify new drivers of liver regeneration [23,24]. In the future, advances in single-cell analytical techniques covering transcriptomics, proteomics, and epigenetic changes will help unravel the complex regenerative microenvironment induced by chronic liver damage and pave the way towards promising therapeutic targets.

## 4. Targeting Key Signaling Pathways and Molecules for Enhancing Liver Regeneration in Disease and Injury

Several key signaling pathways play a critical role in enhancing liver regeneration and promoting the proliferation and differentiation of hepatocytes. Understanding these signaling pathways is crucial for developing therapeutic strategies to enhance liver regeneration in various liver diseases and injuries.

The Wnt/β-catenin pathway is a key signaling pathway in liver regeneration. Wnt ligands bind to frizzled receptors and coreceptors, resulting in the stabilization and deposition of β-catenin in the cytoplasm [25]. This accumulated β-catenin in the cell interacts with transcription factors to regulate gene expression and activate target genes involved in cell-cycle progression. In the liver, Wnt signaling is crucial for tissue regeneration and metabolic zonation and contributes to various liver diseases, including liver cancer [26].

Several regulators control the Wnt pathway, including the two negative regulators called Rnf43 and Znrf3 [27,28], which are responsible for the degradation of frizzled receptors of the pathway. Mutations in these have been found in human cancers, but their role in liver disease is not well understood. In one study, researchers specifically deleted *Rnf43* and *Znrf3* in adult hepatocytes and observed degeneration of the liver, increased unsaturated lipids, altered lipid distribution, and steatohepatitis [29]. The loss of *Rnf43/Znrf3* also affected hepatocyte proliferation. These effects were partially explained by cell-autonomous mechanisms, as liver organoids lacking *Rnf43/Znrf3* showed lipid accumulation and reduced differentiation capacity. These findings are consistent with clinical observations of liver cancer patients with RNF43/ZNRF3 mutations who also exhibit metabolic abnormalities and poorer prognosis.

The Notch signaling pathway, which regulates cell fate during development, has been found to be dysregulated in obese rodents and individuals with NAFLD. In the liver, Notch activation promotes the differentiation of hepatic progenitor cells into cholangiocytes, while inactive Notch signaling supports the development of hepatocytes [30]. To investigate the role of hepatocyte-specific Notch activation in NASH, researchers conducted studies using mice fed with a specific diet that induces NASH and fibrosis [30]. They found that hepatocyte-specific loss of Notch function reduced liver fibrosis without affecting hepatocellular injury or inflammation. Conversely, forced activation of hepatocyte Notch signaling led to fibrosis development. The researchers also identified the involvement of specific genes, sex-determining region Y-box 9 (Sox9) and Sox9-dependent expression encoding the secreted fibrogenic factor Osteopontin (*Spp1*), in the fibrotic process. Furthermore, treating NASH mice with a Notch antagonist resulted in decreased liver fibrosis. Interestingly, it was also observed that Notch activity is increased in hepatocytes of patients with NASH.

In another study, the researchers aimed to investigate the role of highly specialized NPCs of the liver called liver sinusoidal endothelial cells (SECs) in the regeneration process [31]. They specifically investigated the expression and function of c-kit, a stem cell marker and type III receptor tyrosine kinase, in SECs [32]. The study revealed that c-kit was predominantly expressed in SECs and that c-kit^+^ SECs played a crucial role in inducing hepatocyte proliferation following partial hepatectomy (PH) via angiocrine signaling [33]. Its distribution was found to be closely associated with the expression pattern of the liver zonation marker Wnt2, indicating a connection between c-kit^+^ SECs and liver zonation. Mutations in the Notch pathway affected the distribution of c-kit and liver zonation, leading to altered hepatocyte proliferation. Activation of the Notch pathway hindered liver regeneration by inhibiting the positive effects of c-kit^+^ SECs on hepatocytes. Moreover, aside from their involvement in liver regeneration, c-kit^+^ SECs were seen to have therapeutic potential in attenuating liver injury induced by toxins. Infusion of c-kit^+^ SECs in mice protected against liver damage induced by toxins.

These findings suggest that hepatocyte-specific Notch signaling plays a crucial role in the development of the liver by regulating cell fate decisions and tissue homeostasis. Modulating Notch signaling can influence liver regeneration and the balance between liver progenitor cell-mediated repair and hepatocyte proliferation.

Another signaling pathway, Signal Transducer and Activator of Transcription 3 (STAT3), plays a critical role in liver regeneration by mediating the response to various growth factors and cytokines. Binding of these factors to their receptors leads to the activation of their respective intrinsic kinases. For example, as discussed earlier, binding of IL-6 to its receptor on hepatocytes initiates the activation of Janus kinases (JAKs), which leads to the phosphorylation of STAT3 [7]. Phosphorylated STAT3 forms homodimers or heterodimers and translocates to the nucleus, where the dimers bind to specific DNA sequences known as STAT-binding elements (SBEs) in the promoters of target genes [34]. This binding activates transcription of various genes involved in cell proliferation or cell-cycle progression.

Moh et al. investigated the direct effect of STAT3 on liver tissues by generating liver-specific STAT3 knockout mice [35]. Higher mortality rates were observed in the mice less than 24 h after PHx, suggesting that STAT3 is required for survival in the early stages of liver regeneration. Surviving STAT3 knockout mice showed reduced DNA synthesis but were able to restore their liver mass, implying that, while STAT3 may play a role in hepatocyte proliferation, other compensatory mechanisms might also be involved during the process. Carbon tetrachloride-treated STAT3 knockout mice showed increased infiltration of neutrophils and monocytes in the liver, indicating an exaggerated inflammatory response after hepatocyte necrosis, suggesting that STAT3 deficiency might influence immune regulation and contribute to inflammation. In addition to IL6, activation of STAT3 by IL-22 is known to play a role in hepatoprotection. Radaeva et al. demonstrated increased IL-22 expression in T cell-mediated hepatitis. Blocking IL-22 worsened liver injury and reduced STAT3 activation, while administering recombinant IL-22 prevented liver injury, indicating its protective role in hepatocytes [36]. Abdelnabi et al. reported a sex-dependent hepatoprotective role of IL-22 in NAFLD and showed that lack of IL-22 receptor signaling in female mice exacerbated liver injury, apoptosis, inflammation, and liver fibrosis [37].

The Hippo/Yap signaling pathway has emerged as a critical regulator of liver injury and regeneration, playing a significant role in maintaining liver balance and promoting hepatocyte proliferation. Activation of this pathway leads to the phosphorylation and inactivation of the YAP protein, which regulates gene expression and cell proliferation [38]. Conversely, liver-specific deletion of key Hippo pathway kinases results in hepatocyte proliferation because of YAP overexpression [39]. Research has also shed light on the role of Hippo/YAP signaling in liver cell fate determination [40]. Elevated YAP activity is associated with hepatic progenitor identity. When YAP is ectopically activated in differentiated hepatocytes, it causes their de-differentiation, liver overgrowth, and the emergence of oval cells. The Notch signaling pathway has been identified as an important downstream target of YAP in these cells.

Another study found that YAP/TAZ are not crucial for liver development and regeneration in hepatocytes. Instead, they indirectly contribute to liver regeneration by maintaining the integrity of bile ducts and ensuring proper immune cell recruitment and function [41]. The researchers observed that in response to the liver injury caused by carbon tetrachloride, the YAP/TAZ were activated in hepatocytes. However, when *Yap/Taz* were specifically deleted in adult hepatocytes, there was no significant impairment in liver regeneration. On the contrary, when *Yap/Taz* genes were deleted in adult bile ducts, which are the tubes that carry bile in the liver, severe defects and delays in liver regeneration were observed. Further investigation revealed that the mutant bile ducts underwent degeneration, leading to a condition called cholestasis, which is characterized by the buildup of bile in the liver. This cholestasis hindered the recruitment of phagocytic macrophages, which are immune cells that help clear cellular debris from the injury sites. The elevated levels of bile acids activated a transcription factor called the pregnane X receptor, whose activation by an agonist recapitulated the defects observed in the mutant mice.

In addition to the above pathways, other pathways, such as the EGF receptor (EGFR) pathway and the HGF/c-Met pathway, also play a significant role in regulating hepatocyte proliferation, survival, migration, and differentiation during liver regeneration.

In one study, researchers used two methods to investigate the role of a protein called β1-integrin (Itgb1) in liver regeneration: inducible gene deletion using Cre/loxP-mediated gene deletion and nanoparticle-encapsulated small interfering RNA (siRNA) against Itgb1 [42]. By studying genetically modified mice, they were able to understand how specific proteins contribute to the process of liver regeneration. While Itgb1 was shown to be essential for liver regeneration, they also found that Itgb1 cooperates with growth factor signaling components—HGF, c-Met, and EGFR—which are known to play crucial roles in liver regeneration [43,44]. When c-Met or EGFR was deleted in the liver, it led to diminished hepatocyte proliferation and impaired liver regeneration.

In summary, the signaling pathways play a critical role in liver injury and regeneration. They regulate hepatocyte proliferation, survival, and tissue repair in response to liver injury. Dysregulation or impairment of these pathways can lead to impaired liver regeneration, delayed wound healing, or the development of liver diseases, such as fibrosis and cirrhosis (Figure 2). Therefore, targeting these signaling pathways and their components holds great potential for therapeutic interventions to enhance liver regeneration in various liver diseases and injuries. Understanding the intricate mechanisms of these signaling pathways will provide new insights into liver regeneration and may pave the way for novel therapeutic approaches for liver diseases.

While much research has focused on understanding the signals that promote liver regeneration, less is known about the mechanisms that control its termination and the factors that suppress excessive cell growth. These proliferation inhibitors play a crucial role in preventing uncontrolled cell proliferation and the development of liver tumors.

One example of a proliferation inhibitor is peroxisome proliferator-activated receptor-gamma (PPAR-γ), a nuclear receptor that regulates cell-cycle arrest and apoptosis in tumor cells. Research has demonstrated an inverse relationship between PPAR-γ expression and liver regeneration, as its levels decrease shortly after PH but increase during later stages of regeneration. Treatment with pioglitazone, a PPAR-γ agonist, has been shown to suppress liver cell proliferation [45].

Vitamin D3 upregulated protein 1 (VDUP1) is another regulator of cell proliferation that is negatively correlated with liver regeneration. Research has demonstrated that mice lacking VDUP1 show enhanced proliferative responses during liver regeneration, characterized by increased expression of cell-cycle proteins and activation of proliferative signals [46].

Transmembrane and ubiquitin-like domain containing 1 (Tmub1) is another identified gene that has been observed to be upregulated during later stages of liver regeneration. It has a negative effect on hepatocyte proliferation induced by IL-6, suggesting its role in regulating liver cell growth [47]. The expression of Tmub1 may be under the control of IL-6 and CCAAT/enhancer binding protein β (C/EBPβ), which is a crucial transcription factor associated with IL-6 signaling [48].

TGF-β is a pleiotropic cytokine that has both pro-regenerative and anti-regenerative effects on the liver, depending on the context. In the initial phase of liver regeneration, TGF-β signaling suppresses hepatocyte proliferation to prevent excessive cell growth. Early studies demonstrated that TGF-β, derived from platelets, inhibits DNA synthesis in adult rat hepatocytes and suppresses liver regeneration in animal models [5]. Other studies showed TGF-β expression to increase at 4 h and cease at 72 h following PHx, coinciding with the cessation of DNA synthesis, indicating its involvement in the inhibition and termination of liver regeneration [49,50]. Additionally, TGF-β is known to act through EGF and HGF inhibition [49,51], among other mechanisms, and therefore modulating its signaling can help fine-tune the regenerative response in liver diseases and injuries.

These examples highlight the importance of proliferation inhibitors in controlling liver regeneration and preventing excessive cell growth. Understanding the mechanisms by which these inhibitors function can provide valuable insights into the regulation of liver regeneration and the development of potential therapeutic strategies.

## 5. Advancements in the Field: Current Progress and Future Directions

Recent advancements in the field of liver injury and regeneration have shed light on key mechanisms and potential therapeutic approaches. One such approach is hepatocyte transplantation that involves the transplantation of functional hepatocytes into a patient’s liver to replace or supplement the lost or damaged liver function. Hepatocyte transplantation offers advantages over whole organ transplantation, including the ability to keep the native liver intact, and holds promise for treating various liver diseases, such as acute liver failure, and certain metabolic disorders [52]. However, the limited availability of high-quality hepatocytes remains a concern. Long-term engraftment and survival of transplanted hepatocytes remains challenging as transplanted cells experience a decline in functionality and numbers, potentially requiring repeated transplantations. Immune rejection might also lead to reduced survival and functionality of transplanted cells. To overcome this, current research is focused on identifying alternative cell sources for transplantation, including stem cells, fetal hepatoblasts or hepatocytes, and immortalized cells. Efforts to generate and expand human hepatocytes in animals, as seen in humanized liver mice, are ongoing but have met with limitations, such as immune response or the risk of zoonotic diseases. Nevertheless, efforts are aimed to address the shortage of donor hepatocytes and expand the potential applications of hepatocyte transplantation in treating a wide range of liver diseases. Some of the other therapeutic approaches are discussed below.

### 5.1. Gene Editing

Advances in gene therapy and genome editing technologies have offered new avenues for studying liver metabolism and treating liver diseases. Gene editing using CRISPR-Cas9 has emerged as a powerful tool in the field of liver disease and regeneration. CRISPR-Cas9 allows precise modifications of specific genes, enabling researchers to target and edit disease-causing mutations or manipulate genes involved in liver regeneration processes.

In 2014, Xue et al. used hydrodynamic tail vein injection (HDTV) to deliver CRISPR plasmid DNA into the livers of wild-type mice, targeting Phosphatase and tensin homolog (*Pten*) and *p53,* two tumor suppressor genes [53]. Mutating *Pten* led to an increase in Akt phosphorylation and accumulation of lipids in the liver cells, mimicking the effects of deleting the gene using conventional methods. Shortly after, another group of researchers delivered a Cas9 system targeting the *Pten* gene in animal liver using an adenoviral vector [54]. The delivery of Cas9 resulted in efficient gene editing, although the immune responses associated with the adenoviral vector were present. Four months post-treatment, mice that received the *Pten* gene edited adenovial vector showed liver enlargement and features of NASH. These studies provided some of the first human liver disease models in vivo.

Yin et al. used the CRISPR-Cas9 system to correct a mutation in the fumaryl acetoacetate hydrolase (*FAH*) gene in the livers of mice with hereditary tyrosinemia, a genetic disease [55]. They delivered the CRISPR-Cas9 components to the liver using HDTV, resulting in the expression of the wild-type Fah protein in a small percentage of liver cells. The expansion of these corrected cells led to the rescue of the bodyweight loss associated with the disease. In another study, researchers used a combination of lipid nanoparticles (LNPs) and adeno-associated viruses to deliver Cas9 mRNA, guide RNA (sgRNA), and a repair template DNA to correct mutation in the *FAH* gene [56]. One study identified four novel liver tumor suppressor genes, *Nf1*, *Plxnb1*, *Flrt2*, and *B9d1*, by conducting a genome-wide CRISPR/Cas9-based knockout screen targeting 20,611 genes in mice using liver progenitor cells that overexpressed Myc and lacked p53 [57].

Despite its immense potential, gene editing using CRISPR-Cas9 faces challenges. Efficient delivery of CRISPR components to target liver cells remains a hurdle, requiring the development of safe and effective delivery methods. Off-target effects and immune responses to CRISPR components are other considerations that need to be addressed to ensure the safety and long-term efficacy of gene editing therapies.

### 5.2. Stem Cells

Stem cells offer a remarkable capacity for self-renewal and differentiation into diverse cell types, making them a promising avenue for liver disease treatment. Different types of stem cells, such as mesenchymal stem/stromal cells (MSCs), hematopoietic stem/progenitor cells (HSPCs), and induced pluripotent stem cells (iPSCs), have been extensively studied for their regenerative abilities. These stem cells possess self-renewal and differentiation capabilities and contribute towards tissue repair and regeneration.

MSCs are cells with a fibroblast-like appearance that possess the ability to differentiate into various cell types, including hepatocyte-like cells (HLCs). They have the capability to secrete cytokines, chemokines, and growth factors which support tissue repair and regeneration and exhibit immunomodulatory properties [58,59]. Liu et al. investigated their effect on liver regeneration and explored the underlying mechanisms. The findings revealed that the infusion of MSCs improved liver regeneration by enhancing cell proliferation and growth in the early stages after PHx [60]. Additionally, MSCs restored liver synthesis function and reduced lipid accumulation through the mechanistic target of the rapamycin (mTOR) signaling pathway. The infusion of MSCs also promoted a proinflammatory environment characterized by increased expression of IL-6 and IL-1β, activated the STAT3 and Hippo-YAP pathways, and consequently led to enhanced cell proliferation. The researchers concluded that MSCs enhanced liver function and facilitated liver regeneration after extensive resection through paracrine mechanisms. These observations indicate that MSCs hold promise as a potential therapeutic approach for treating acute liver failure following hepatectomy.

In addition to their differentiation potential, factors such as conditioned medium derived from MSC cultures and human umbilical cord MSC-derived exosomes (hUCMSC-EVs) have been shown to promote liver regeneration in a rat model of hepatic ischemia-reperfusion injury by modulating inflammatory responses, reducing apoptosis and reducing oxidative stress [61,62]. In addition, extracellular vesicles from a newly derived stem cell population called human liver stem cells (HLSCs) have been shown to contribute towards tissue repair and regeneration [63]. HLSCs were first identified in 2006 through a unique cultivation method that involved stringent culture conditions. This technique allowed mature hepatocytes to undergo cell death, leaving behind expandable clones of HLSCs that exhibited remarkable multipotent capabilities [64]. HLSCs can further differentiate into a variety of cell types, including hepatocytes, endothelial cells, and islet-like cell organoids, under controlled laboratory conditions. In addition, HLSCs express markers associated with both MSCs and hepatocytes, indicating a partial commitment toward hepatic lineage. Importantly, HLSCs exhibit immunomodulatory properties, inhibiting the activation of immune cells such as T-lymphocytes, natural killer cells, and dendritic cells [63]. While HLSCs hold significant potential for various therapeutic applications, their full potential is still being explored. By leveraging the immunomodulatory and regenerative properties of HLSCs, research could pave the way for novel treatments that could significantly impact the field of liver disease management.

Hematopoietic stem/progenitor cells (HSPCs), abundant in the bone marrow, have also been investigated for their potential in liver regeneration. Transplantation of HSPCs into fumarylacetoacetate hydrolase (Fah)-deficient mice showed that they could transdifferentiate into functional hepatocytes, leading to the regeneration of the injured liver, thus correcting the phenotype in a mouse model of human tyrosinemia type I [65].

Induced pluripotent stem cells (iPSCs), generated by reprogramming adult cells, have the ability to differentiate into various cell types, including HLCs. They offer advantages such as expandability, bankability, and reproducibility. iPSC-derived HLCs have been used for disease modeling and have shown potential for liver regeneration in animal models [59]. In addition, patient-specific iPSCs can also be generated, avoiding issues of immune rejection, and therefore hold great potential for regenerative medicine and personalized therapies.

The use of stem cells represents a promising avenue for liver disease and regeneration. By harnessing the regenerative potential of stem cells, promoting endogenous repair mechanisms, and delivering therapeutic factors, stem-cell-based approaches have the potential to revolutionize the treatment of liver diseases, improve patient outcomes, and reduce the need for liver transplantation.

### 5.3. Biomaterials and Tissue Engineering

The use of stem cells in combination with tissue engineering approaches, such as hydrogels and scaffolds, holds further promise for enhancing liver regeneration and transplantation outcomes.

Scaffold-based liver tissue engineering involves using specialized structures made from natural or synthetic materials. These structures, called scaffolds, are designed to mimic the ECM of the liver. The ECM provides a supportive environment for cells by promoting attachment, growth, and specialization [66]. Decellularized liver scaffolds, created by removing cellular components while preserving the ECM, can be re-seeded with live cells to regenerate liver tissue [66]. Various cell sources can be used for recellularization, including primary hepatocytes, iPSCs, and MSCs. The use of scaffold-based approaches has shown promise in improving liver function and promoting tissue regeneration in animal models. Previously, the therapeutic potential of extracellular vesicles derived from mesenchymal stem cells (MSC-EVs) for liver diseases was discussed. However, their effectiveness is hindered by the rapid clearance of MSC-EVs from the liver, limiting their impact. To address this challenge, researchers developed a sustained release approach utilizing clickable polyethylene glycol (PEG) hydrogels [67]. MSC-EVs were encapsulated within these hydrogels, allowing for gradual release over a period of one month. In a rat model of chronic liver fibrosis, the sustained release of MSC-EVs from the hydrogels demonstrated superior antifibrosis, anti-apoptosis, and regenerative effects when compared to the conventional injection of MSC-EVs without hydrogel encapsulation. This sustained delivery strategy extends the availability and therapeutic benefits of MSC-EVs in the context of chronic liver failure.

Three-dimensional bioprinting is an advanced technique in tissue engineering that enables the precise fabrication of 3D structures using live cells [68]. Bioprinting allows for the creation of biomimetic liver tissue that closely mimics the microenvironment of the liver [69]. Bioprinted liver tissues have demonstrated the ability to recapitulate drug-induced fibrogenesis and show phenotypic and functional enhancements of cells [70]. Bioprinting has also been used to generate liver organoids for transplantation [71].

However, despite the advancements, challenges remain. Proper vascularization of engineered tissues is crucial for long-term survival and function. Complex vascular networks, mimicking the natural anatomy of the liver, are still difficult to fabricate using current bioprinting techniques. Strategies such as incorporating angiogenesis growth factors, coculturing endothelial cells, and creating microchannels have been explored to improve vascularization [72].

### 5.4. Liver Organoids

Liver organoids or three-dimensional cell culture systems that mimic the structural and functional attributes of the liver have emerged as valuable tools for studying hepatic biology, disease modeling, and regenerative medicine [73]. Liver organoids can be generated from various cell sources, such as iPSCs and embryonic stem cells (ESCs), and provide more physiologically relevant and scalable models compared to traditional two-dimensional cell cultures. Tsuchida et al. demonstrated a safe and effective treatment for chronic liver damage in rats by transplanting liver organoids into the liver through the portal vein. The method helped to regenerate damaged livers and showed promising results in improving liver function and survival rates [74]. Organoids derived from pluripotent stem cells have also been used to model various liver diseases, such as steatosis and steatohepatitis. In one study, researchers successfully co-differentiated epithelial and stromal lineages from pluripotent stem cells to create multicellular liver organoids [75]. These organoids, when treated with free fatty acids, demonstrate the progressive development of pathology similar to steatohepatitis, involving accumulation of fat, inflammation, and fibrosis. The stiffness of the organoids could also be used as a biophysical readout to assess fibrosis severity. This organoid culture system provided a robust platform for modeling complex liver diseases and drug screening. In another study, researchers successfully differentiated organoids into functional hepatocytes and cholangiocytes [76]. When exposed to free fatty acids, the hepatic organoids exhibited gene expression patterns resembling those found in liver tissues of patients with NASH. Furthermore, incubation with free fatty acids resulted in structural alterations characteristic of NASH. This hepatic organoid platform provided a valuable tool for modeling complex liver diseases.

Liver organoids offer several advantages, including disease modeling, drug testing, and developmental studies. However, like any other experimental model, liver organoids have their limitations, such as capturing the full complexity of the native organ, achieving fully mature hepatocytes, and challenges, such as the long-term viability and functionality of organoids, exist [77,78].

In addition, while liver organoids offer valuable insights into liver biology, they often do not accurately replicate the complex architecture of the liver lobule due to their limited vascularization [79]. Liver organoids provide a more uniform microenvironment compared to the distinct gradients seen in the native liver lobule. This uniformity fails to mimic the molecular gradients that drive zonal gene expression. The spatial arrangement of cell types in liver organoids also differs from that in the native liver, which affects cellular interactions and signaling gradients critical for zonal differentiation.

Liver organoids also show limitations in capturing the interactions that extend beyond the liver itself. The broader systemic interactions, such as those with the intestinal barrier, the nervous system, and the immune system, cannot be fully replicated in an isolated in vitro model. Researchers studying liver diseases and treatments should therefore complement organoid studies with in vivo and ex vivo models to gain a more comprehensive understanding of liver pathology.

Despite these limitations, liver organoids still offer valuable insights into liver biology and disease, and ongoing research is focused on improving their functionality and relevance for various applications. Strategies to enhance vascularization, cellular composition, and spatial organization within organoids are actively being pursued to address these limitations and make them more representative of the native liver’s architecture and functions.

In conclusion, liver organoids are in vitro models that replicate in part the complexity and functionality of the liver tissue. They offer a powerful platform for personalized medicine and hold great potential for disease-specific therapeutic strategies. Recognizing their advantages and disadvantages is crucial for effectively interpreting experimental results, and continued research should be aimed at addressing these limitations and improving the utility of liver organoids for various applications.

### 5.5. siRNA-Based Therapeutics

The adequate understanding of the intrinsic molecular pathogenesis of liver disorders has also led to a surge in research efforts towards altering expression levels of specific genes involved in the pathophysiology of various diseases using RNA interference (RNAi). In addition, there is a growing pool of oligonucleotide-based therapies in clinical trials that have shown promising potential against various diseases. Oligonucleotide-based therapeutics include siRNA, anti-miRs, miRNA mimics, and antisense oligonucleotides. These therapeutics modulate gene expression, provide high-level specificity, and reduce off-target effects.

siRNA-based therapeutics have gained significant attention in recent years, and several siRNA drugs have been approved for specific indications due to the recent advancement in the development of N-acetylgalactosamine (GalNAc) siRNA conjugates. The GalNAc moiety acts as a ligand and binds to a receptor called asialoglycoprotein (ASGPR) expressed on hepatocytes [80]. GalNAc-siRNA conjugates can be injected subcutaneously, efficiently taken up by liver cells, and subsequently released to induce a therapeutic effect.

Onpattro™ (Patisiran) was the first approved siRNA-based therapeutic, used for hereditary polyneuropathy [81]. It was formulated as a lipid nanoparticle (LNP) that serves as a versatile platform for delivering therapeutic molecules, including nucleic acids and drugs, and is delivered directly to liver cells. Vutrisiran (HELIOS-A), a second-generation siRNA drug, is now approved for the same indication. Vutrisiran targets the same mRNA as Onpattro but is conjugated with a GalNAc molecule which enhances its stability [82]. Some of the other GalNAc-siRNA conjugates approved are Givlaari (givosiran) for acute hepatic porphyria [83], Leqvio (inclisiran) for adults with hypercholesterolemia or mixed dyslipidemia [84], and Olpasiran, which targets apolipoprotein A [85].

While siRNA-based therapeutics are showing promising results in preclinical and clinical studies, there are still challenges to overcome. These include improving the stability, specificity, and delivery efficiency of siRNA molecules, as well as addressing potential off-target effects and immune responses. To overcome the limitation of siRNA delivery, researchers used gold nanoparticles modified with branched polyethyleneimine (PEI), a cationic polymer, for the delivery of siRNA against the *c-Myc* gene, which is overexpressed in HCC [86]. The nanoparticles showed high cellular uptake, no significant toxicity, and successful delivery of siRNA to cancer cells, resulting in significant gene silencing.

Another method utilized a nanocomplex to deliver siRNA to liver cells. This complex consisted of protamine, which neutralizes the anticoagulant effect of heparin, and siRNA targeting AKT. The complex was enveloped by a conjugate of hyaluronic acid and taurocholic acid (HA-TCA). This conjugation assisted the complex in penetrating the cells and escaping from endosomes, protecting the siRNA from the harsh gastric environment [87]. The complex effectively reached liver cancer cells due to the recycling system of enterohepatic bile acids induced by TCA. The controlled release of siRNA was achieved through the degradation of the conjugate by the enzyme hyaluronidase present in cancer cells. The uptake by hyaluronic acid receptors in liver cancer cells led to the retardation of cancer cell growth and reduction in tumor size in a murine model of colorectal liver metastasis.

A recent study studied the therapeutic potential of targeting c-Jun N-terminal kinase-2 (Jnk2) in chronic liver disease (CLD) and end-stage liver cancer [88]. A hepatocyte-specific lipid-based siRNA formulation called siJnk2, using the LNP system, was developed. Treatment with siJnk2 resulted in reduced apoptotic cell death and attenuated hepatocarcinogenesis. siJnk2 treatment also led to decreased fibrogenesis, ameliorated markers of hepatic damage, and reduced the formation of premalignant nodules, suggesting a potential therapeutic effect in inhibiting tumor initiation.

Recently, STP707, an siRNA-based therapy, received FDA approval to proceed with clinical trials for primary sclerosing cholangitis (PSC), a chronic liver disease characterized by inflammation of the bile ducts (https://clinicaltrials.gov/ct2/show/NCT03841448, accessed on 19 July 2023). STP707 is based on a combination therapy, specifically targeting two key molecules, TGF-β1 and COX-2, known to be overexpressed in different liver cells, including Kupffer cells and liver SECs. Intravenous administration of STP707 successfully reduced the expression of the targeted genes in preclinical studies, exhibited a good safety profile, and demonstrated anti-fibrotic activity.

These studies highlight different approaches for delivering siRNA to target specific genes involved in liver diseases, with promising outcomes in gene silencing and inhibition of cancer progression. Ongoing research and advancements in siRNA design as well as delivery systems hold great potential in developing siRNA-based therapeutics for various diseases.

### 5.6. Single-Cell Transcriptomics

Single-cell transcriptomics is a powerful technique that has revolutionized our understanding of the liver’s architecture at a cellular level. Advancements in single-cell RNA sequencing (scRNA-seq) technology has provided unprecedented insights into the cellular composition, cell states, and intercellular communication within the liver microenvironment.

scRNA-seq has enabled the discovery of previously uncharacterized cell types and states in the liver. MacParland et al. obtained liver tissues from five healthy donors and analyzed the transcriptional profiles of 8444 individual liver cells. The researchers not only identified 20 distinct cell populations within the liver but also discovered two different populations of the macrophages of the liver along with distinct functions, thereby providing an in-depth view at a cellular level [89]. Ramachandran et al. profiled transcriptomics of over 100,000 human liver cells from healthy and cirrhotic individuals, identified different cell types present in the liver, and uncovered novel subpopulations of macrophages (characterized by the expression of TREM2 and CD9) and endothelial cells (expressed ACKR1 and PLVAP) that are associated with liver fibrosis, thus enabling the discovery of therapeutic targets [90].

scRNA-seq coupled with computational analyses has allowed the study of intercellular communication and signaling networks within the liver microenvironment during injury and regeneration. By conducting ligand–receptor modeling, researchers investigated the interactions between scar-associated macrophages, endothelial cells, and collagen-producing mesenchymal cells. They identified several signaling pathways, including TNF receptor superfamily member 12A (TNFRSF12A), platelet-derived growth factor (PDGFR), and Notch signaling, that were involved in promoting fibrosis within the scarred areas of the liver [90]. This provided insights into the crosstalk between different cell populations and the signaling pathways involved in liver regeneration.

In another study, researchers aimed to understand how heterotypic interactions in 3D organoids affect lineage identity during liver development [91]. They used scRNA-seq to analyze the gene expression pattern of human liver cells in both 2D culture and 3D liver bud organoids. They reconstructed the lineage progression of HLCs from pluripotent stem cells in 2D culture and observed the emergence of heterogeneity during hepatoblast differentiation. They then compared the 3D liver bud organoids to fetal and adult human liver cells and uncovered a striking similarity between the organoids and fetal liver cells. Lastly, using receptor–ligand pairing analysis and inhibitor assays, the researchers investigated the signaling pathways involved in liver bud development and found vascular endothelial growth factor (VEGF) signaling to promote endothelial network formation and hepatoblast differentiation in the organoids. The study provides insights into the cellular and molecular processes involved in liver development and the influence of cell–cell interactions on lineage identity. The utilization of 3D liver bud organoids provides an invaluable model system for investigating liver development and holds promise in advancing the development of regenerative therapies for liver diseases.

Single-cell transcriptomics has emerged as a powerful technique for dissecting cellular heterogeneity and molecular dynamics. Researchers employed scRNA-seq to examine the heterogeneity of BECs and hepatocytes in healthy and injured livers [92]. Researchers found significant heterogeneity in homeostatic BECs, which was associated with the activation of a YAP-dependent program. This dynamic cellular state was responsive to injury and played a role in BEC survival and hepatocyte reprogramming into biliary progenitors. The findings highlight the molecular heterogeneity within the ductal epithelium of the liver and emphasize the regulatory role of YAP in liver regeneration.

Single-cell transcriptomics has provided a comprehensive understanding of the gene regulatory networks that orchestrate tissue repair and regeneration following liver injury. To understand the dynamics of liver regeneration, researchers used spatially resolved scRNA-seq to investigate the regeneration process in mouse liver after acute acetaminophen (APAP) intoxication [93]. The study revealed that hepatocytes across the liver lobule engaged in proliferation, exerting the necessary mitotic pressure to swiftly replenish the damaged pericentral zone (a central region surrounding the central vein that plays a critical role in various metabolic functions). During the regeneration process, a specific subset of hepatocytes at the regenerating front exhibited transient upregulation of fetal-specific genes as they underwent reprogramming into a pericentral state. Moreover, distinct cell populations, including endothelial cells, hepatic stellate cells, and macrophages, demonstrated specific or zone-specific roles in immune recruitment, proliferation, and matrix remodeling throughout the regeneration phase. These findings provide valuable insights into the coordinated programs involved in zonal liver regeneration.

Another group delved into the regenerative potential of the adult liver and sought to unravel the underlying mechanisms that enable it to restore both mass and function following injury. They utilized a PHx mouse model and employed scRNA-seq as well as a single-cell assay for transposase accessible chromatin sequencing (scATAC-Seq) analyses on approximately 13,000 individual hepatocytes [94]. The study uncovered that, following PHx, the hepatocytes exhibited diversification into multiple distinct populations characterized by different functional attributes. Some hepatocytes retained the chromatin landscapes and transcriptomes akin to undamaged adult liver cells, while others underwent a transition, acquiring fetal-like characteristics that rendered them more proliferative and instrumental in the regeneration process. Additionally, the research shed light on the heterogeneity and dynamic nature of hepatocyte responses during liver regeneration. Despite maximal proliferative activity, a significant portion of hepatocytes retained the chromatin landscape and metabolic functions of healthy hepatocytes, indicating their commitment to maintaining essential liver-specific metabolic responsibilities. On the other hand, a larger population of hepatocytes exhibited changes in chromatin accessibility and gene expression associated with liver development and regeneration, suggesting a transition towards a more proliferative state. The researchers also demonstrated that epigenetic mechanisms play a crucial role in orchestrating and coordinating these complex cell state transitions during liver regeneration. Overall, the study provides insights into the mechanisms underlying liver regeneration and the balance between increased proliferative activity and the maintenance of vital liver-specific functions. Understanding the factors and mechanisms that drive these cellular transitions will be crucial for understanding defective liver repair, liver failure, and carcinogenesis and may contribute to the development of novel approaches for preventing and treating liver-related conditions.

Overall, scRNA-seq has emerged as a powerful tool for dissecting the cellular heterogeneity, lineage trajectories, and molecular mechanisms underlying liver injury and regeneration. It has the potential to drive the development of novel therapeutic strategies and personalized medicine approaches for liver diseases. Additionally, the development of advanced computational tools and algorithms will be crucial for the analysis and interpretation of large-scale scRNA-seq datasets, enabling the identification of rare cell populations and complex cellular states.

## 6. Challenges and Opportunities in Developing Effective Therapies for Liver Disease: A Roadmap for Future Research

Developing effective therapies for liver disease comes with various challenges. There is a gap between animal models and clinical studies when it comes to understanding liver regeneration and translating that knowledge into therapeutic benefit. Animal models have been valuable in uncovering the mechanisms of liver regeneration, the signaling pathways involved, the timing of the regenerative response, and the cellular sources of regenerative cells. On the other hand, clinical studies in humans have focused more on observing clinical outcomes and identifying factors associated with the outcomes. Bridging the gap between preclinical studies and clinical translation is crucial. Conducting rigorous preclinical studies; utilizing humanized animal models, organoids, and bioengineered liver models to better mimic human liver physiology and disease; and developing robust translational strategies to accelerate the clinical translation of liver diseases and improve the predictability of clinical outcomes are some of the strategies that should be employed at a consensus level [16].

Additionally, one of the primary challenges in developing therapies for liver disease is the complexity and heterogeneity of liver pathologies. Different liver diseases can arise from various etiologies, and each disease presents unique pathological features. For example, various approaches have been proposed to target NASH pathogenesis. These include modulating inflammation, enhancing fatty acid metabolism, inhibiting de novo lipogenesis, preventing hepatocyte injury, and investigating antifibrotic therapies [95]. Drugs targeting fibroblast growth factor (FGF) analogs, nuclear receptors (PPARs and FXR), thyromimetics (synthetic analogs of thyroid hormones with tissue-specific thyroid hormone actions), ASK1 inhibitors, caspase inhibitors, and CCR2/5 antagonists have been explored. However, treating NASH has proven to be challenging, as many agents in clinical trials have failed to meet their primary endpoints.

Various drugs targeting different pathways, such as PPAR agonists (elafibrinor and seladelpar), FGF analogs (aldafermin and pegbelfermin), apoptosis inhibitors (selonsertib and emricasan), and the CCR2/CCR5 inhibitor cenicriviroc, did not achieve the desired outcomes in terms of NASH resolution, fibrosis improvement, or hepatic fat reduction [96,97,98]. These results were observed in phase IIb and III trials and highlight the difficulties in developing effective therapies. Nevertheless, new molecules continue to be tested in clinical trials, and different approaches have emerged, such as the use of combination therapy.

Combining monotherapies with distinct mechanisms of action that act synergistically and target multiple pathways simultaneously might be more effective in circumventing compensatory mechanisms or cross-reactivity. Combination therapies involving different agents, such as FXR-agonists, CCR2/5 inhibitors, antidiabetic agents, and acetyl-CoA carboxylase (ACC) inhibitors, are currently being investigated in phase II trials [95].

One such combination involves cilofexor, an FXR agonist, and firsocostat, an ACC inhibitor, which has demonstrated promising results in reducing liver fat and improving liver enzymes in NASH patients (https://classic.clinicaltrials.gov/ct2/show/NCT02781584, accessed on 19 July 2023). Additionally, combination therapies involving antidiabetic drugs, such as GLP-1 agonists (https://classic.clinicaltrials.gov/ct2/show/NCT03987074, accessed on 19 July 2023) and SGLT2 inhibitors (https://classic.clinicaltrials.gov/ct2/show/NCT04065841, accessed on 19 July 2023), are being investigated for their potential to enhance outcomes related to both liver health and diabetes. Combination therapy may also help in mitigating side effects associated with individual drugs, such as the use of statins to mitigate LDL cholesterol increase caused by FXR agonists (https://clinicaltrials.gov/ct2/show/NCT02633956, accessed on 19 July 2023).

Despite the advancements, selecting the right therapeutic target and the right combination of drugs is crucial. scRNA-seq has shown that different subsets of immune cells have different functions in NASH, indicating that neutralizing or inhibiting entire cell types may not be suitable. A better understanding of the pathogenesis of NASH can aid in the development of more targeted therapies. Liver fibrosis can develop as a consequence of advanced NASH. Currently, there is no standard therapy for liver fibrosis, and early detection is challenging. Activated HSCs play a crucial role in liver fibrogenesis, and targeting these cells is important for effective treatment. Various protein markers, such as type VI collagen receptor, retinol-binding protein receptor, PDGFR, and others, have been identified to be overexpressed in activated HSCs. However, delivering therapeutic agents to activated HSCs remains challenging due to their low abundance and barriers in the fibrotic liver [3].

Efforts have been made to develop targeted and enhanced delivery systems, including small molecules, antibodies, proteins, lipids, and nucleic acids. Researchers are also exploring affinity selection technologies to discover peptide- or antibody-based ligands with higher affinity and flexibility for chemical modifications [99]. These ligands, which can be artificially designed, have several advantages. They are small in size, making them easier to produce, and they do not elicit an immune response. Peptides and aptamers are examples of such ligands that can be used in targeted delivery systems [99]. Recently Vitamin A-coupled LNP (https://classic.clinicaltrials.gov/ct2/show/NCT02227459, accessed on 19 July 2023) has been utilized for moderate to extensive hepatic fibrosis. Incorporating HSC-specific ligands into antifibrotic agents has the potential to significantly improve the success rate of clinical studies.

Developing technologies such as cell therapy or oligonucleotide-based therapy may hold potential for treating liver diseases. As highlighted earlier, cell therapy using stem cells has emerged as a promising alternative for the treatment of liver fibrosis and cirrhosis, with MSCs being the most commonly used cell type. MSCs derived from different tissues, such as bone marrow and umbilical cord, have been investigated, and bone-marrow-derived MSCs have shown superior improvement in liver function parameters compared to umbilical cord-derived MSCs [100]. However, in cell therapy clinical trials, the number of cells, the route of administration, and the cell type are crucial considerations. Additionally, the cell count used in treatment depends on factors such as patient weight, clinical condition, and administration route [100]. The route of administration often involves infusing the stem cells via the hepatic artery to enhance engraftment. Safety and efficacy are equally crucial, and although clinical trials have demonstrated improvements in liver function parameters without significant adverse effects, further research and larger-scale studies are needed to optimize cell therapy protocols and fully understand the long-term effects of stem-cell-based therapies for liver diseases.

In addition, cell-free therapy utilizing secreted factors and EVs by MSCs is an emerging strategy that avoids potential risks associated with cell transplantation, such as tumorigenicity and embolism [100]. However, the standardized extraction of large quantities of EVs and exosomes from stem cells remains a challenge, and their optimal dosage and half-life are not well-defined. Therefore, clinical trials involving stem-cell-derived EVs for the treatment of liver fibrosis and cirrhosis are still limited.

While there have been significant advancements in oligonucleotide-based therapies for liver diseases, challenges remain in the safe and effective intracellular delivery of these compounds as well as their intracellular processing. One of the challenges in siRNA delivery is the efficient release of siRNAs from endosomes/lysosomes upon cellular internalization. Currently, only a small fraction of internalized siRNAs escapes the endosomal/lysosomal system, highlighting the need for alternative strategies to enhance their release [101]. While LNPs have been successful in liver and solid tumor targeting, their size limits their extravasation from the bloodstream, potentially restricting their use in other diseases [102]. Other concerns with LNPs include the need for intravenous administration and the potential toxicity of excipients. Nevertheless, the progress in oligonucleotide therapies, both in preclinical and clinical stages, has paved the way for innovative therapies. Despite the challenges, the exploration of siRNA drugs beyond the liver and the potential of siRNA combinations or other non-coding RNAs are areas that offer considerable opportunity.

## 7. Conclusions

In conclusion, developing effective therapies for liver disease is a complex and multifaceted endeavor. Addressing the challenges associated with disease heterogeneity, impaired liver regeneration, and lack of specific biomarkers is critical. Advancements in precision medicine and personalized therapy based on genomic and transcriptomic data integration offer opportunities in the treatment of liver diseases. By considering individual patient characteristics, including genetic variations and biomarkers, treatments can be tailored to achieve better outcomes. Through collaborative efforts and leveraging emerging technologies, we can make significant strides towards combating liver diseases and improving the lives of millions of individuals affected by these conditions.

## Figures and Tables

**Figure 1 cells-12-02129-f001:**
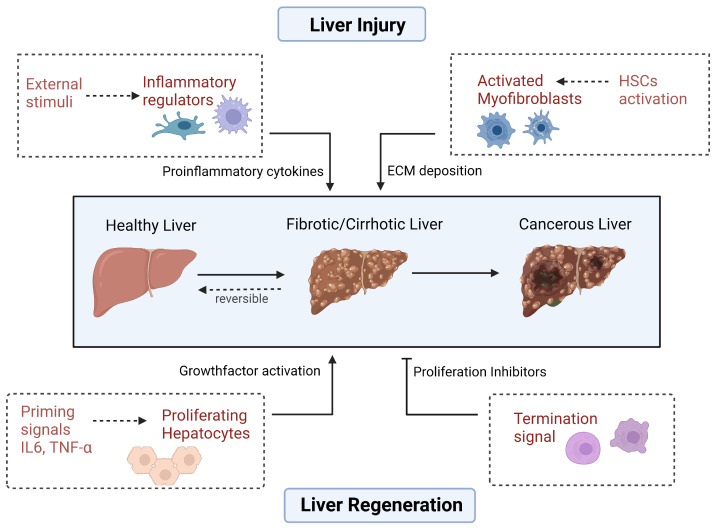
Various factors influence liver homeostasis. Liver injury triggers the activation of various pathway components that play a critical role in acute vs. chronic liver damage. Liver injury also initiates compensatory mechanisms for tissue regeneration that involve various growth factors and proliferation inhibitors that contribute to the initiation and controlled proliferation of hepatocytes. Created with BioRender.com.

**Figure 2 cells-12-02129-f002:**
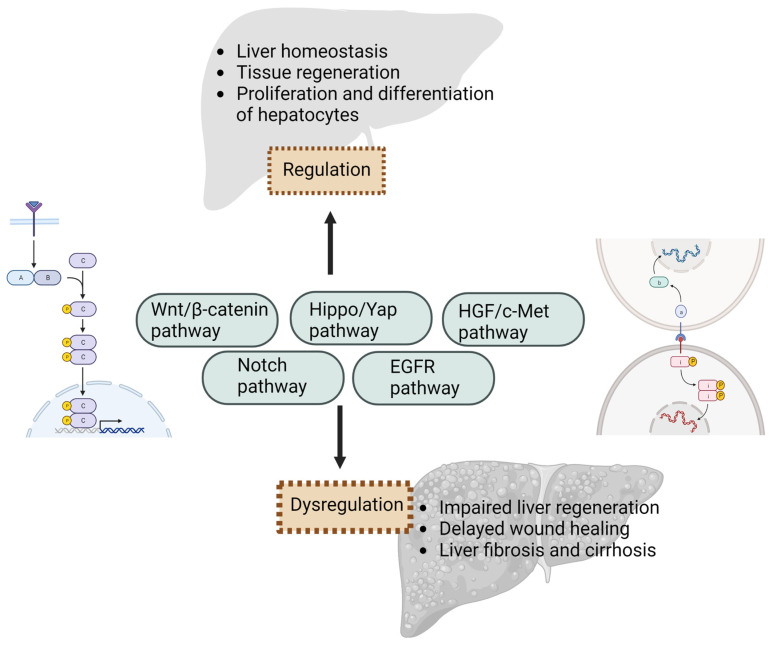
Signaling pathways play a critical role in liver injury and regeneration. Multiple signaling molecules act in concert to regulate hepatocyte proliferation, survival, and tissue repair in response to liver injury. However, dysregulation or impairment of these pathways lead to impaired liver regeneration, delayed wound healing, or liver diseases, such as fibrosis and cirrhosis. Created with BioRender.com.

## Data Availability

Not applicable.

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
