# Peer review of "Liver Injury and Regeneration: Current Understanding, New Approaches, and Future Perspectives"

_cells, 2023, doi:10.3390/cells12172129_

Round 1

Reviewer 1 Report

In this manuscript Hora & Wuestefeld provide an interesting review on liver injury/regeneration with a valuable overview on current developments and future perspectives. 

The following comments and suggestions are made by this reviewer. 

1)  line 48/49: 

‘… by engaging germline-encoded pattern-recognition receptors (PRRs) such as monocytes, macrophages, neutrophils, epithelial cells and adaptive immune system cells.’

PRRs are receptors not cell types.

2) line 139/140:

As discussed earlier, pro-inflammatory signals as TNF- α and IL-1β are important to trigger hepatocytes to enter the cell cycle and proliferate.’

Authors correctly emphasize the fact that pro-inflammatory cytokines like TNFa or IL-1b play an important role in initiating liver regeneration. However, molecular mechanisms behind that are not delineated. Authors should briefly provide more detailed information at that point e.g. by describing the role of NF-kB in hepatocyte cell fate decisions (proliferation versus apoptosis).

3) At some positions, references are missing and should be added.

For example, line 293-298:

‘Other studies showed TGF-. expression increase at 4 hr and cease at 72 hr following PHx, coinciding with the cessation of DNA synthesis, indicating its involvement in the inhibition and termination of liver regeneration. Additionally, TGF-. is known to act through EGF and HGF inhibition, among others and therefore modulating its signaling can help fine-tune the regenerative response in liver diseases and injuries.’

References backing that section are lacking.

4) Section 5.4.

This reviewer shares the enthusiasm concerning liver organoids. However, limitations should be discussed: 

For example, organoids do not reflect the architecture of the liver lobule and thus do not reflect e.g. zonal gene expression. Other key issues in liver pathology that are not covered by cultivating organoids include interactions with distant organs (e.g. the intestinal barrier) or the nervous system (particularly the vagus nerve) as well as the spatiotemporal influx and function of specific leukocyte populations. 

Those limiting aspects should be discussed in balanced manner.

5) line 355/356:

Authors only very briefly touch on the role of STAT3 in liver regeneration (in the context of IL-6). Given its pivotal role, STAT3 should be discussed in some more detail – this should also include IL-22, which is therapeutically more relevant, as compared to IL-6.  Of note, IL-22-based biologicals are currently in clinical trials.

Reviewer 2 Report

This reviewer appreciates the thoroughness of the review, which focuses on liver regeneration. the manuscript takes into consideration both the cellular and molecular mechanisms involved in liver damage and subsequent regeneration, and addresses the issue of predictiveness of both animal and cellular liver injury models, also indicating how all these studies offer the possibility of identifying potential targets molecules to improve liver regeneration.

Some criticisms of the text concern paragraph 5:

- in section 5.2 (stem cells) speaking of MSCs the EVs are described and it is said (lines 365-367) that also the EVs from HLSCs have shown regenerative capacity. However, these cells are not introduced and are only mentioned for the EVs. If possible, please add a couple of lines describing these cells and their use in vitro and/or in vivo

- section 5.5 (single cell transcriptomics) is very interesting and offers food for thought demonstrating how new technologies and in silico analyzes can help and be exploited to try to better understand a possible approach to be used to promote liver regeneration . However, this part compared to the others in chapter 5, does not concern a therapeutic approach. This reviewer suggests moving this part (5.5) either to the end of chapter 5 as a further possible approach or to chapter 4 indicating how single cell transcriptomics can help pinpoint key signaling pathways

- In general, it would be helpful to add some information about the state of the art of hepatocytes transplantation , maybe before the 5.2 stem cells

- line 318, 320 and 321: please change "adenovirus vector" with "adenoviral vector" or "adenovirus-derived vector"

check for any typos or spelling errors (e.g. line 76: signals such such as interleukin 6; line 369-370: Transplantation of HSPCs into fumarylacetoacetate hydrolase (Fah)-deficient mice showed it could transdifferentiate into functional hepatocytes...since the sentence is referring to HSPCs "it" should be replaced with "they" or "these cells") 

check the abbreviations (e.g. HDTV: only the acronym appears in line 313 and in line 325, it is never written in full)

Round 2

Reviewer 1 Report

All points raised have been addressed adequately.